# SPIRE: Synergistic Planning, Imitation, and Reinforcement for Long-Horizon Manipulation

**Zihan Zhou**[1,2], **Animesh Garg**[1,3], **Dieter Fox**[1], **Caelan Garrett**[1*], **Ajay Mandlekar**[1*]

[1] NVIDIA, [2] University of Toronto, Vector Institute [3] Georgia Institute of Technology, [*] equal contribution

**Abstract:** Robot learning has proven to be a general and effective approach for programming manipulators. Imitation learning is able to teach robots solely from human demonstrations but is bottlenecked by the capabilities of the demonstrations. Reinforcement learning uses exploration to discover better behaviors; however, the space of possible improvements can be too large to start from scratch. And for both approaches, the learning difficulty increases exponentially to the length of the manipulation task. Accounting for this, we propose SPIRE, a system that first uses Task and Motion Planning (TAMP) to decompose tasks into smaller learning subproblems and second combines imitation and reinforcement learning to maximize their strengths. We develop novel strategies to train learning agents when deployed in the context of a planning system. We evaluate SPIRE on a suite of long-horizon and contact-rich robot manipulation problems. We find that SPIRE outperforms prior approaches that integrate imitation learning, reinforcement learning, and planning by 35% to 50% in average task performance, is 6 times more data efficient in the number of human demonstrations needed to train proficient agents, and learns to complete tasks nearly twice as efficiently. View https://sites.google.com/view/spire-corl-2024 for more details.

**Keywords:** Reinforcement Learning, Manipulation Planning, Imitation Learning

## 1 Introduction

Reinforcement Learning (RL) is a powerful tool that has been widely deployed to solve robot manipulation tasks [1, 2, 3, 4]. The RL trial-and-error process allows an agent to automatically discover solutions to a task and improve its behavior over time. However, in practice, it often relies on careful reward engineering to guide the exploration process [5, 6]. The exploration burden and reward engineering problem is even more challenging to overcome for long-horizon tasks, where an agent must complete several subtasks in sequence in order to solve the task [7].

Imitation Learning (IL) from human demonstrations [8, 9] is a popular alternative to reinforcement learning. Here, humans teleoperate robot arms to collect task demonstrations. Then, policies are trained using the data. This alleviates the burden of reward engineering, since correct behaviors are directly specified through demonstrations. This paradigm has recently been scaled up by collecting large datasets with teams of human operators and robots and shown to be effective for different real-world manipulation tasks [10, 11, 12]. While these agents can be effective, they typically are imperfect, with respect to both success rates and control cost, and not robust to different deployment conditions, especially when it comes to long-horizon tasks [13].

One way to integrate the benefits of both IL and RL is to first train an agent with IL and then finetune it with RL. This can help improve the IL agent and make it robust through trial-and-error , while also alleviating the need for reward engineering due to the presence of the demonstrations. Several works have used this paradigm successfully, but long-horizon manipulation still remains challenging due to the burden of exploration and long-term credit assignment [7].

8th Conference on Robot Learning (CoRL 2024), Munich, Germany.

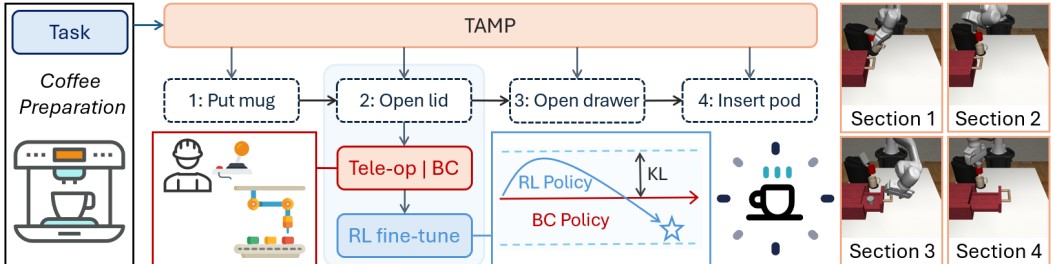

Figure 1: **SPIRE Overview.** (**Left**) SPIRE first attempts to solve the task with a TAMP system. When the TAMP planner encounters an action deemed too hard to plan, it then enters the *handoff* section and delegates the action to a human teleoperator to manually complete it. We record the trajectories from the human operators to build a demonstration dataset and train an IL policy with it. Finally, we train an RL policy to fine-tune the IL policy via warmstarting and deviation constraining. (**Right**) The four handoff sections in *Coffee Preparation*.

One effective approach for long-horizon manipulation is to leverage a hybrid control paradigm, where the agent is only responsible for local manipulation skills, instead of the full task [14, 15]. An example is the HITL-TAMP system [14], where an agent is trained with IL on small contact-rich segments of each tasks, and the rest of the task is performed using Task and Motion Planning (TAMP) [16]. Another related approach is PSL [15], which learns an agent using RL instead of IL on small segments, and uses motion planning to sequence the learned skills together. These approaches are effective for challenging long-horizon manipulation tasks, but they still often do not train perfect policies, suffering from some of the pitfalls of IL and RL. In this paper, we take inspiration from these approaches and create a hybrid control learning framework that allows for efficient imitation learning and RL-based finetuning of agents to address long-horizon manipulation tasks.

We introduce Synergistic Planning Imitation and REinforcement (SPIRE), a system for solving challenging long-horizon manipulation tasks through efficient imitation learning and RL-based fine-tuning. SPIRE decomposes each task into local manipulation segments that are learned with a policy and segments handled by a TAMP system. The manipulation segments are first trained via imitation learning and then finetuned with reinforcement learning. Our approach on 9 challenging manipulation tasks reaches an average success rate of 87.8%, vastly outperforms TAMP-gated IL [14] (52.9%) and RL [15] (37.6%). In the subset of tasks where SPIRE and IL both reach a high success rate, our method only uses 59% of the steps required by IL to complete the task. In *Tool Hang*, SPIRE fine-tunes an IL policy with only 10% success rate to 94%. We perform a thorough analysis of our method and also show that in many cases, a handful of demonstrations suffice for learning challenging tasks. Compared with IL, SPIRE improves the overall demo efficiency by 5.8 times in the evaluated subset of tasks.

**Our contributions are as follows:**
• We propose SPIRE, a hybrid learning-planning system that synergistically integrates the strengths of behavior cloning, reinforcement learning, and manipulation planning. SPIRE first learns a TAMP-gated policy with BC and then improves it with RL.
• We introduce key insights to enable RL-based finetuning with sparse rewards in this regime, including a mechanism to warmstart the RL process using the trained BC policy, a way to constrain exploration to be close to the BC agent behavior, and a multi-worker TAMP framework to optimize the throughput of SPIRE's RL process.
• We evaluate SPIRE on a suite of long-horizon contact-rich tasks and find that it outperforms prior hybrid learning-planning approaches **success rate averaged across tasks** (87.8%, compared to 52.9% and 37.6%), **execution efficiency** (episodes are only 59% the length of the BC agent), and **human demonstration efficiency** (6 times less data required than BC to train similar agents).

## 2 Related Work

**Hierarchical approaches for long-horizon tasks.** Hierarchical approaches decompose the challenging long-horizon tasks into easier-to-solve subtasks. RL based methods explore the division of

sub-tasks with reusable skills [17, 18, 19, 20, 21]. [22, 23, 24, 25, 26] build hierarchical RL solutions with subpolicies and metacontrollers. Our work instead leverages a planner that provides guidance on which policies to learn as well as initial and terminal state distributions of tasks, compared to bottom-up HiRL methods, which tend to be data inefficient. Notably this top-down breakdown may also be achieved with a Language Model which can provide a plan composed of steps and sub-goal targets [27, 28, 29, 30, 31, 15].

**Robot manipulation with demonstrations.** Behavior cloning (BC) [32] learns a policy by directly mapping observations to actions and is typically trained end-to-end using pre-collected pairs of observation and behavior data. While this is seemingly a supervised learning problem, the context of robotics adds challenges. BC datasets tend to contain data sampled from *multimodal distributions*, due to intra-expert variations. Recent work address this problem with implicit models including those derived from energy-based models [33, 34], diffusion models [35, 36, 37, 38, 39], and transformer models [40, 41, 42]. Another challenge is the *correlation in sequential data*, which can lead to policies which are susceptible to temporally correlated confounders [43]. Recently several works have set out to handle this by predicting action chunks. For example, the Action Chunking Transformer (ACT) line of work [44, 45] and diffusion policy [35]. While BC-based methods combined with high-capacity models enable complex robotics tasks from demonstrations, problems in robustness and long-horizon generalization remain.

Experts and their demonstrations can be used in combination with RL in multiple ways, including acting as task specifications, exploration guidance, and data augmentation. Inverse RL [46, 47, 48] learns a reward model for RL from demonstrations; [49, 50, 51] explore using demonstrations to bootstrap the reinforcement learning process; [52] uses state matching for reward computation in RL. [53] shares a similar setup with ours, where they also warmstart RL with a BC policy and use a masked BC loss the constrain the RL policy from deviating. [54, 55] propose to fine-tune a semi-expert initial policy by training a residual policy on top of it with RL. Different from the mentioned works, we focus on multi-stage robotic manipulation tasks with high-dimensional input spaces.

**Learning for Task and Motion Planning.** Task and Motion Planning (TAMP) is a search and planning-based approach for sequential manipulation tasks [16, 56, 57, 58]. However, pure TAMP-based methods suffer from reliance on accurate environment modeling and prior knowledge of the skills, making those methods less suitable for tasks involving complex contact-rich skills.

# 3 Method

Our approach Synergistic Planning Imitation and REinforcement (SPIRE) learns and deploys closed-loop visuomotor skills within a TAMP system (see Fig. 1). First we frame our problem as a policy learning problem across a sequence of Markov Decision Processes (Sec. 3.1). Next, we describe our approach for incorporating both classical and learned robot skills into TAMP (Sec. 3.2) to enable TAMP-gated learning. Next, we describe how we train an initial agent with TAMP-gated Behavioral Cloning (BC) (Sec. 3.3). We then propose an RL-based finetuning algorithm to improve the BC agent with RL (Sec. 3.4). Finally, we introduce a parallelized training scheduler that is able to intelligently manage dependencies among stages when conducting RL in our setting (Sec. 3.5).

## 3.1 Problem Formulation

In our setup, each robot manipulation task can be decomposed into a series of alternating TAMP sections and *handoff sections*, where TAMP delegates control to a trained agent $\pi$. These sections are *TAMP-gated* [14], as they are chosen at the discretion of the TAMP system, and typically involve skills that are difficult to automate with model-based planning. We wish to train an agent $\pi$ to complete these handoff sections efficiently and reliably. We model our TAMP-gated policy learning problem as a series of Markov Decision Processes (MDPs), $\mathcal{M} := (\mathcal{S}, \mathcal{A}, T, \{r^i\}, \{p_0^i\}, \gamma)_{i=1}^N$, where $N$ is the number of MDPs (each corresponds to a TAMP handoff section), $\mathcal{S}$ and $\mathcal{A}$ are the state and action space, $T$ is the transition dynamics, $r^i(s)$ and $p_0^i$ are the $i$-th reward function and initial state distribution, and $\gamma$ is the discount factor. The start and end of each handoff section is chosen by TAMP, consequently, TAMP determines the initial state distribution $p_0^i$ for each handoff

section, and the reward function $r^i(s)$, which is a sparse 0-1 success reward based on the successful completion of the section. Our goal is to train a stochastic policy $\pi : \mathcal{S} \rightarrow \Delta_\mathcal{A}$ that maximizes the expected return $J(\pi) := \mathbb{E}_{(i,\tau) \sim \pi} \left[ \sum_{t=0} \gamma^t r_t^i \right]$. We next describe the TAMP system.

## 3.2 TAMP with Learned Skills

Task and Motion Planning (TAMP) [16] is a model-based approach for synthesizing long-horizon robot behavior. TAMP integrates discrete (symbolic) planning with continuous (motion) planning to plan hybrid discrete-continuous manipulation actions. Essential to TAMP is a model of the actions that a planner can apply and how these actions modify the current state. From such a model, TAMP solvers can search over the space of plans to find a sequence of actions and their associated parameters that satisfies the task.

In SPIRE, we seek to implicitly learn a select set of TAMP actions that are impractical to manually model and then combine them with traditional actions through planning. In essence, our strategy is to learn policies $\pi$ that control the system from TAMP *precondition* states to *postcondition* states (typically described by *effects*). We adopt the modeling strategy introduced by Mandlekar et al. [14] and deploy PDDLStream [58] to solve each TAMP problem. See Appendix G for a summary of our planning model. Fig. 2 visualizes an interleaved execution of TAMP trajectory control and RL-learned policy control in the *Tool Hang* domain. Here, the planning model explicitly models the `pick` as well as intermediate `move` actions but defers the `insert` and `hang` actions to the RL agent.

Algorithm 1 describes the SPIRE policy at test time. It begins by observing the state. If the state satisfies the task's goal conditions, SPIRE terminates successfully. Otherwise, SPIRE invokes TAMP to plan a sequence of traditional and learned actions. SPIRE executes the trajectory associated with each traditional action until it reaches the first learned action. At that time, SPIRE executes the closed-loop policy associated with the learned action until it achieves its subgoal condition. To account for the stochastic outcome of the policy, SPIRE replans and repeats this process.

## 3.3 TAMP-Gated Imitation Learning

**TAMP-Gated Data Collection.** We collect an initial dataset of human demonstrations through TAMP-gated human tele-operation, where the human operator collects demonstrations for handoff sections when prompted by TAMP, to form the demonstration dataset $\mathcal{D} = \{\{(s_t, a_t)_{t=1}^{H_i}, g_i\}\}$, where $s_t \in \mathcal{S}$, $a_t \in \mathcal{A}$ and $H_i$ is the horizon, and $g_i$ is the handoff section of the $i$-th trajectory. To improve the data collection efficiency, we replicate the task queuing system from [14].

---

**Algorithm 1** SPIRE

1: **procedure** SPIRE($G$)
2:     **while True do**
3:         $s \leftarrow$ OBSERVE()
4:         **if** $s \in G$ **then**
5:             **return True**
6:         $\vec{a} \leftarrow$ PLAN-TAMP($s, G$)
7:         **for** $a \in \vec{a}$ **do**
8:             **if** $a.type =$ "RL" **then**
9:                 $\pi \leftarrow a.policy$
10:                 EXECUTE-POLICY($\pi$)
11:                 **break**
12:             **else**
13:                 $\tau \leftarrow a.trajectory$
14:                 EXECUTE-TRAJECTORY($\tau$)

---

**TAMP-Gated Behavioral Cloning.** Given the dataset $\mathcal{D}$, we train a Behavioral Cloning (BC) policy parameterized by $\phi$ to minimize the negative log-likelihood loss over the demonstration dataset: $\phi^* = \operatorname{argmin}_\phi \mathbb{E}_{(s,a) \sim \mathcal{D}}[-\log \pi_\phi(a|s)]$. The trained BC agent $\pi_\phi$ may have substantial room for improvement, depending on the complexity of the task, and the number of demonstrations available for training. We next describe our RL-based finetuning procedure (Sec. 3.4) that allows this agent to be improved through reinforcement learning.

## 3.4 RL Finetuning

Given a trained BC agent $\pi_\phi$, we wish to train an RL agent $\pi_\theta$ to improve performance further. To avoid reward engineering, we only assume access to sparse 0-1 completion rewards for each handoff section provided by TAMP (Sec. 3.1). However, exploration in sparse-reward settings has been shown to be challenging [59, 60, 61, 62], especially in continuous state and action spaces. Fortunately, we can use the BC policy trained in the previous section as a reference point for exploration – we want to restrict the behavior of the RL policy to be in a neighborhood of the BC policy. This is achieved by a) warmstarting the RL policy optimization using the BC policy, and b) enforcing a constraint on the deviation between the RL policy and the BC policy.

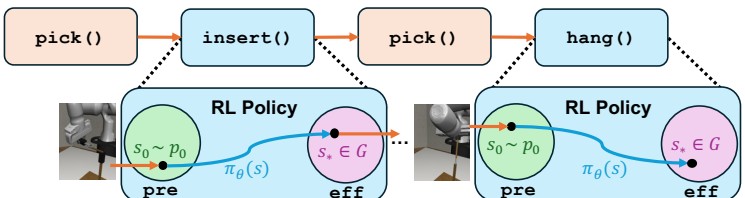

**Figure 2: SPIRE execution.** SPIRE computes a TAMP plan but defers execution of certain contact-rich skills, such as `insert` and `hang`, to learned agents – we call these handoff sections. The preconditions of each handoff section define the initial state distribution of the agent, and the postconditions of each action correspond to the termination states of the corresponding MDP for the handoff section.

**Warmstarting RL optimization with BC.** We tested two ways to warmstart the RL agent. *Initialization.* One method is to initialize the weights of the RL agent with those of the trained BC agent, $\theta \leftarrow \phi^*$, where $\phi^* = \arg\min_\phi L_{BC}(\phi)$, and subsequently finetune the weights with online RL objectives. Despite being easy to implement, this can be less flexible since it requires the agent structure of the RL and BC policies to match. Furthermore, researchers have found that retraining neural networks with different objectives can cause the network to lose plasticity [63], which can make the policy harder to optimize because of the objective shift from BC to RL. *Residual Policy.* An alternative way is to fix the BC policy as a reference policy and train a residual policy on top of it. Let the residual policy be $\pi_\theta^+(s)$. The residual policy shares the same action space as the normal policy but is initialized to close to zero. The final action is defined as a summation of the reference action $a \sim \pi_{\phi^*}(s)$ and the residual action $a^+ \sim \pi_\theta^+(s)$. In practice, we only add the mean of the reference policy to the residual action instead of sampling the reference action.

**Constraining Deviation between BC and RL agents.** The sparsity of reward signals produces high-variance optimization objectives, which can lead the RL policy to quickly drift away from BC and lose the exploration bonus from warmstarting. Therefore, it is critical to constrain the policy output to be close to the BC agent throughout the training process. We achieve this by imposing a KL-divergence penalty. We conclude our RL optimization objective as follows: $J_{FT}(\theta) := J(\pi_\theta) - \alpha D_{KL}(\pi_\theta \| \pi_{\phi^*})$, where $D_{KL}(p \| q) := \mathbb{E}_{(s,a)\sim p}\left[\log \frac{p(a|s)}{q(a|s)}\right]$ and $\alpha$ is the weight for the penalty term.

### 3.5 Multi-Worker Scheduling Framework

Making our TAMP-gated framework compatible with modern reinforcement learning procedures requires addressing several challenges. First, TAMP can take dozens of seconds for a single rollout, which severely lowers the throughput of RL exploration. Second, the TAMP pipeline executes each section sequentially, which means that later handoff segments can only be sampled when previous handoff segments are completed successfully. This leads to an imbalance of episodes for the different handoff segments and is potentially problematic for the RL agent. We propose a multi-worker TAMP scheduling framework to integrate TAMP into RL fine-tuning. The framework consists of three components – a group of TAMP workers that run planning in parallel, a status pool that stores the progress of the workers, and a scheduler that distributes tasks to the workers and balances the initial states. We further describe how the framework allows for curriculum learning, and how the framework accelerates learning efficiency for RL training. See Appendix H for more details.

**TAMP workers.** Each TAMP worker has an environment instance and repeatedly runs a TAMP planner. Upon reset, the TAMP worker initiates TAMP until a handoff section has been reached. It then sends a pair (`#worker`, `#section`) representing its ID and which handoff section it has entered to a FIFO status queue, indicating that it is ready to take RL agent actions. The worker then enters an idle state until it receives a command from

---

**Algorithm 2** Scheduler

1: **procedure** SCHEDULER(WORKERS, STATUSQUEUE, POLICY, STRATEGY)
2:     **while** True **do**
3:         $i, j \leftarrow$ STATUSQUEUE.$pop()$
4:         **if** STRATEGY.$accepts(j)$ **then**
5:             **while not** WORKERS[$i$].$done()$ **do**
6:                 $s_{obs} \leftarrow$ WORKERS[$i$].$observe()$
7:                 $a \leftarrow$ POLICY.$act(s_{obs})$
8:                 WORKERS[$i$].$step(a)$
9:         **else**
10:             WORKERS[$i$].$reset()$

---

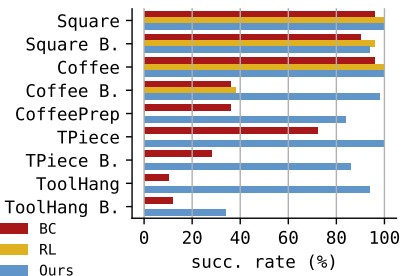

| Episode Duration: | BC [14] | RL [15] | Ours |
|---|---|---|---|
| Square | 18.1 | **8.3** | 11.6 |
| Square Broad | 24.5 | **8.4** | 13.6 |
| Coffee | 63.1 | **15.0** | 38.4 |
| Coffee Broad | 80.6 | **25.7** | 61.3 |
| Coffee Preparation | 193.3 | - | **168.5** |
| Three Piece | 58.7 | - | **34.0** |
| Three Piece Broad | 62.2 | - | **38.1** |
| Tool Hang | 81.8 | - | **61.7** |
| Tool Hang Broad | 130.5 | - | **109.8** |

**Figure 3: Full evaluation.** Comparing the success rates (**left**) and the average duration (**right**) of successful rollouts of *HITL-TAMP-BC* (**BC**), *TAMP-gated Plan-Seq-Learn* (**RL**), and SPIRE (**Ours**) across all 9 tasks. Each datapoint is chosen from the best run out of 5 seeds and is averaged from 50 rollouts. SPIRE improves the BC policy in terms of both success rate and average duration in all 9 tasks and reaches 80% success rate in 8. RL has an advantage in average duration in the easier set of tasks but fails to learn anything in the rest.

the scheduler. Depending on the command, the worker either resets itself or starts interacting with the environment by exchanging actions and states with the scheduler. If the current section is solved, the worker sends a success signal to the scheduler and runs TAMP until the next handoff section.

**Scheduler.** (Algorithm 2) The scheduler is a centralized component that manages the TAMP workers. It also provides an environment abstraction to the single-threaded RL process. The scheduler is configured with a sampling strategy. Upon initialization, it first pops an item from the status queue. According to the sampling strategy, the scheduler either rejects this section, in which case it sends a resetting signal to the corresponding worker; or starts a new episode and interacts with the worker.

**Curriculum Learning.** The behavior of the scheduler depends on a sampling strategy, allowing it to function as a *curriculum* for the RL agent. We consider two strategies: `permissive` is the default strategy that allows all sections through, while `sequential` only accepts a section when the success rate of passing all the previous sections reaches a threshold. `sequential` allows controlling the initial state distribution during the early stages of training, to ensure the RL agent achieves proficience in each section sequentially before continuing onto the next section.

**Remarks on Efficiency.** Suppose a TAMP planning process takes at most $T$ seconds over the episode; each environment interaction step, counting communication latency, takes at least $t$ seconds; and each handoff section is at least $H$ steps. If the number of TAMP workers $n \geq \frac{T}{tH}$, the proposed multi-worker TAMP scheduling framework reaches a throughput of at least $1/t$ frames per second. In comparison, the single-worker counterpart has a worst-case throughput of $\frac{H}{T+tH}$ frames per second. Suppose that the planning process is slower than the handoff sections by a factor $k$ (e.g. $T = k \cdot tH$), then our framework is faster than the single-worker alternative by a factor of $k + 1$.

## 4 Experiments

**Tasks.** For evaluation, we follow [14] and choose a set of long-horizon manipulation tasks, namely *Square*, *Coffee*, *Three Piece*, and *Tool Hang*. We also include the broad variants of those tasks, where we use a broad object initialization region, and *Coffee Preparation*, which has the longest horizon with four handoff sections. See Appendix E for more details.

**Environment Details.** *Observation space.* For most tasks, we use a single $84 \times 84$ RGB image from the wrist-view camera. For *Tool Hang*, we use the front-view camera instead since the wrist-view is mostly occluded. For *Tool Hang Broad* and *Coffee Preparation*, we use both wrist-view and front-view cameras, as well as proprioception state (end-effector pose and gripper finger width). *Action space.* Actions are 7-dimensional (3-dim delta end-effector position, 3-dim delta end-effector rotation, 1-dim gripper actuation). *Horizon.* Each handoff section is limited to 100 steps (5 seconds with 20Hz control frequency) for all tasks, except for *Tool Hang Broad*, where the limit is 200 steps.

**Baselines.** We compare our method with two baselines: *HITL-TAMP-BC* (**BC**), which is adapted from [14] to match our network structure; and *TAMP-gated Plan-Seq-Learn* (**RL**), which is adapted

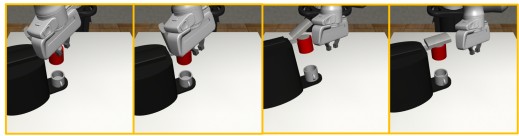 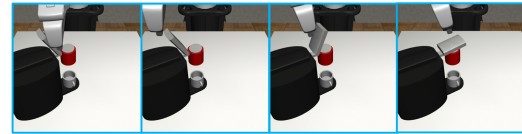

**(a)** Naive RL agent          **(b)** SPIRE agent

**Figure 4: Qualitative comparison.** Rollouts of vanilla RL vs our method. The first agent attempts to close the lid by knocking the coffee machine, while our agent follows the demonstrations and closes the lid with fingers.

from [15] by replacing the LLM-based planning system with our TAMP system for fair comparison. We collected 200 human demonstrations for each task to train the behavior cloning policy. For RL, we use DrQ-v2 [64] as the base algorithm. See Appendix B for more details.

**Evaluation.** We evaluate each trained agent for 50 rollouts and report the success rate and average completion steps in the successful rollouts. We train 5 seeds for each algorithm and report the best-performing agent (success rate-wise, tie-breaking with average steps) unless otherwise specified.

## 4.1 Results

**SPIRE outperforms both TAMP-gated BC and RL.** We compare our method with the TAMP-gated BC [14] and RL [15] baselines across all 9 tasks (see Fig. 3). SPIRE reaches 80% success rate in 8 out of 9 tasks, while BC and RL only reach 80% in 3 tasks each. In *Tool Hang*, our method reaches 94% success rate despite the BC counterpart only having 10%, which is over 9-times improvement. Remarkably, this low-performing BC agent is enough to help address the exploration burden (unlike RL, 0% success) and train a near-perfect agent. Across all 9 tasks, SPIRE averages a 87.8% success rate, while BC and RL only average 52.9% and 37.6% respectively.

**SPIRE produces more efficient agents than BC through RL fine-tuning.** SPIRE agents have lower average completion times than their BC and RL counterparts (Fig. 3, right). Even in tasks such as *Square*, *Square Broad*, *Coffee*, *Three Piece*, where BC policies already have high success rates, our method improves the efficiency by only using an average of 59% completion time.

**SPIRE's use of the BC agent helps address the RL exploration burden on challenging long-horizon tasks.** Exploration in RL with sparse rewards is extremely challenging, especially for robot manipulation tasks for their continuous and high-dimensional observation and action space. Our method solves the initial exploration problem by anchoring policy learning around the BC agent. As shown in Figure 3, RL policies without utilizing BC only reach nonzero success rates in *Square*, *Coffee* and their *Broad* variants, all of which have only one handoff section and relatively shorter horizons. Even in *Coffee Broad*, RL encounters exploration difficulties due to the broader object distribution, resulting in only partially solving the task.

**Qualitatively, SPIRE can improve agent behavior without introducing undesirable behavior, unlike RL.** Safety awareness has always been a critical matter in robotics learning. Safety constraints can be hard to define with numerical values, which adds to the challenges of realizing safety in RL. We notice that in *Coffee*, RL policy has a much shorter completion time than our method. This is at the cost of ignoring safety concerns. We compare two rollouts of RL and our method in Figure 4. The RL-trained policy attempts to close the lid by knocking the coffee machine with the arm, which can potentially damage the robot and the coffee machine and even cause danger to humans; while our method preserves safety awareness by following the demonstration's practice of closing the lid with its fingers.

**SPIRE can train proficient agents using just a handful of human demonstrations.** BC methods can require several human demonstrations to train proficient agents, which can be a major drawback due to the cost of collecting this data [9]. We reduce the number of human demonstrations used by SPIRE to 10 and 50 (instead of 200 as in Fig. 3), and we plot the minimum of demonstrations needed to reach at least 80% success rate in Fig. 5. As the plot shows, SPIRE can successfully fine-tune a BC policy trained with as few as 10 demos in all evaluated tasks except for *Tool Hang*

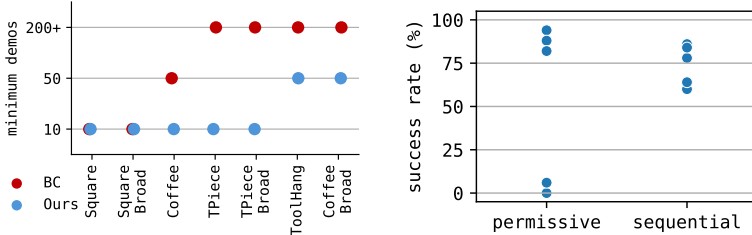

**Figure 5: Demo efficiency and sampling strategy ablation.** (**Left**) Minimum number of demos needed to reach at least $80\%$ success rate. (**Right**) Success rates across 5 seeds in *Tool Hang*, comparing `permissive` and `sequential` strategies. `sequential` has a lower variance but `permissive` has the better top-1 policy.

and *Coffee Broad*, for which 50 demos are enough. In the 7 tasks, our method needs 150 demos in total, while BC needs more than 870, a $5.8\times$ improvement in efficiency.

## 4.2 Ablation Study

We conduct two ablative studies to investigate (1) the value of the KL-divergence penalty and (2) the value of curriculum learning, governed by the two scheduler sampling strategies `permissive` and `sequential`. In this section, we compare the performance distribution of the 5 runs instead of only the top-1 run for a more comprehensive evaluation.

**Value of divergence penalty.** We ablate the divergence penalty on two representative tasks, *Three Piece* & *Tool Hang* (Table 6) and observe a drastic performance drop (84% to 17.6%, 74% to 0%).

The sparsity in rewards leads to high-variance optimization objectives for RL. As a result, even when warmstarted with BC, the RL policy can quickly deviate from it, especially when the chance of reaching the reward signal is low. Therefore, constraining the policy close to BC throughout the training is critical. We select two representative tasks, *Three Piece* and *Tool Hang* for this ablation. The result is shown in Figure 6. Without the divergence penalty, the RL policies deviated immediately and never returned.

| Task | w/ (%) | w/o (%) |
|---|---|---|
| Three Piece | **84.0** (**34.7**) | 17.6 (39.4) |
| Tool Hang | **74.4** (**11.8**) | 0.0 (0.0) |

**Figure 6: KL-divergence penalty ablation.** Mean and standard deviation (in parenthesis) of success rates across the 5 seeds in *Three Piece* and *Tool Hang*, with and without KL-divergence penalty. In both tasks, the divergence penalty improves the performance by a large margin.

**Value of curriculum learning.** We compare the two sampling strategies in *Tool Hang* task. The result is shown in Figure 5. `sequential` strategy shows a much smaller variance, while `permissive` produces the better top-1 seed performance. The main difference between the two strategies is how the second section states emerge during training. For `permissive`, the second section states emerge gradually as the success rate of passing the first section gets higher, resulting in a more gentle distributional shift that leads to a higher overall success rate; for `sequential`, the shift is more abrupt, but it has fewer distraction states in the early stage, resulting in a more stable training process.

## 5 Conclusion

We presented SPIRE, an integrated approach for deploying RL, BC, and planning harmoniously. We showed how BC can be used to not only warm-start RL but also guide the RL process via focused exploration. We introduced a scheduling mechanism to improve RL data throughput and increase learning efficiency. Finally, we evaluated SPIRE in simulation against recent hybrid learning-planning baselines and found that SPIRE results in more successful and efficient policies.

**Limitations.** We focus on tasks that center around object-centric manipulation of rigid objects in table-top environments. We assume that a human teleoperator can demonstrate the learned skills to warmstart RL. The TAMP component assumes that the state is observable and comprised of rigid objects, possibly connected with articulation. To simplify RL training, we only considered Markovian policies; however, using neural network architectures with history, such as RNNs, may boost performance [9]. Finetuning BC policy with RL requires the simulation to be efficient.

**Acknowledgments**

The authors thank Vector Institute for computing.

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

# A Overview

The Appendix contains the following content.

- **Policy Learning Details** (Appendix B)): details on hyperparameters used
- **Ablation: SPIRE without TAMP** (Appendix C): ablation study on the effect of removing TAMP-gating and directly running BC and RL fine-tuning
- **Comparison to Additional Methods** (Appendix D): comparison to other RL methods that leverage demonstrations
- **Tasks** (Appendix E): details on tasks used to evaluate SPIRE
- **Variance Across Seeds** (Appendix F): discussion on the variance of results across different seeds and how results are presented
- **TAMP Formulation**(Appendix G): details on the TAMP planner
- **Bridging TAMP Planner and RL** (Appendix H): details on how we integrate the TAMP planner with RL
- **Ablation: SPIRE without Multi-Worker** (Appendix I): ablation study on the effect of using multiple parallelized TAMP workers
- **Additional Experiment Results** (Appendix J): additional experiment results, including RL learning curves

# B  Policy Learning Details

**Table 1:** DrQ-v2 hyperparameters.

| | |
|---|---|
| Network structure | CNN |
| Learning rate | 1e-4 |
| Discount | 0.99 |
| Batch size | 256 |
| $n$-step returns | 3 |
| Action repeat | 1 |
| Seed frames | 4000 |
| Feature dim | 50 |
| Hidden dim | 1024 |
| Optimizer | Adam |

**Hyperparameters**. The base RL algorithm for all our experiments is DrQ-v2 [64]. The specific hyperparameters are in Table 1.

**Observation**. For most tasks, we use one $84 \times 84$ RGB image from the wrist camera as the only observation. For *Tool Hang*, we use a front-view camera instead since the wrist-view is heavily occluded. For *Tool Hang Broad* and *Coffee Preparation*, we use both camera views plus proprioception state (end-effector pose and gripper finger width). We use the default CNN structure from DrQ-v2 to encode the image observations. For tasks with multiple observations, we first encode the image observations each with an independent CNN network, then concatenate the CNN outputs alongside the low-dimensional observations such as proprioception states to form the feature vector.

**Action**. All of our tasks share a 7-dimensional continuous action space. It is models 6-DOF delta movement of the end-effector along with 1 dimension for finger control. The action is modeled as a normal distribution with a scheduled standard deviation.

# C  Ablation: SPIRE without TAMP

We provide an additional ablation study on the high-level planner, TAMP. To do so, we treat the whole task as one handoff section. The agent only receives a reward of one if it completes the whole task. We collect 200 full demonstrations in *Square*, train a BC policy, and apply SPIRE to fine-tune the BC policy. Since the trajectory becomes longer and the robot now needs to handle object transportation, a single local wrist-view becomes insufficient. We thus include both the wrist view and the global front view, as well as the robot proprioception states in the observation for the w/o TAMP variant. The result is shown in Table 2.

**Table 2:** Comparing the success rates of *Square* and *Square Broad* with and without TAMP.

| Task | BC | RL | Ours |
|------|-----|-----|------|
| Square *w/* TAMP | 98% | 100% | 100% |
| Square *w/o* TAMP | 2% | 0% | 94% |
| Square Broad *w/* TAMP | 100% | 100% | 100% |
| Square Broad *w/o* TAMP | 0% | 0% | 0% |

Even though the w/o TAMP variant has more information from observations, the BC and RL policies are significantly worse than the w/ TAMP counterpart. The increased horizon makes the BC policy easier to drift away to regions less frequently visited in demonstrations and makes RL exploration much harder. In *Square*, despite the low starting quality, SPIRE still fine-tunes BC to reach a 94% success rate, demonstrating the effectiveness of RL fine-tuning. However, when the initialization range increases in *Square Broad*, even SPIRE fails to find an acceptable policy.

In conclusion, TAMP (1) confines the agent-controlled section to a small local area, reducing the need for global information, and (2) decreases the horizon (11.6 w/ TAMP, 101.7 w/o TAMP in *Square*) for the learned agent, reducing compounding errors and exploration difficulty.

## D Comparison to Additional Methods

In each handoff section from TAMP, SPIRE utilizes the demonstrations by training a behavior cloning agent and using RL to fine-tune it. There are alternative methods to combine expert demonstrations and RL, which can be readily plugged in as replacements to SPIRE. In this section, we make connections from our method to GAIL [46]. The discriminator-based IRL reward in GAIL serves the same purpose as our KL penalty term - preventing the current policy from deviating from the expert policy. We draw further connection by showing that our KL penalty is the same as the IRL reward function in GAIL with an alternative discriminator objective and a different reward form.

Let $\pi_E$ be the expert policy. The IRL reward function in GAIL is $-\log(1 - D(s, a))$, where $D : \mathcal{S} \times \mathcal{A} \to [0, 1]$ is the discriminator that maximizes

$$J(D) := \mathbb{E}_{\tau \sim \pi}[\log(1 - D(s, a))] + \mathbb{E}_{\tau \sim \pi_E}[\log(D(s, a))]$$

If we use an alternative objective:

$$\hat{J}(D) := \mathbb{E}_{s \sim \pi_E, a \sim \text{Unif}}[-D(s, a)] + \mathbb{E}_{\tau \sim \pi_E}[\log(D(s, a))]$$

The alternative objective discriminates $\pi_E$ from a fixed policy rather than the current learned policy $\pi$. Assume $\pi_E$ has full support, then maximizing $\hat{J}(D)$ is equivalent to maximize for every $s \in \mathcal{S}$:

$$\hat{J}_s(D) := \mathbb{E}_{a \sim \text{Unif}}[-D(s, a)] + \mathbb{E}_{a \sim \pi_E(\cdot | s)}[\log(D(s, a))] \tag{1}$$

$$= -\left(\int D(s, a)\, \mathrm{d}a\right) + \left(\int \pi_E(a \mid s) \log(D(s, a))\, \mathrm{d}a\right) \tag{2}$$

$$= -\left(\int D(s, a)\, \mathrm{d}a\right) + \left(\int \pi_E(a \mid s) \log \pi_E(a \mid s)\, \mathrm{d}a\right) + \left(\int \pi_E(a \mid s) \log \frac{D(s, a)}{\pi_E(a \mid s)}\, \mathrm{d}a\right) \tag{3}$$

$$= -\left(\int D(s, a)\, \mathrm{d}a\right) + H(\pi_E(\cdot \mid s)) + \left(\int \pi_E(a \mid s) \log \frac{D(s, a)}{\pi_E(a \mid s)}\, \mathrm{d}a\right) \tag{4}$$

$$\leq -\left(\int D(s, a)\, \mathrm{d}a\right) + H(\pi_E(\cdot \mid s)) + \left(\int \pi_E(a \mid s)\left(\frac{D(s, a)}{\pi_E(a \mid s)} - 1\right) \mathrm{d}a\right) \tag{5}$$

$$= -\left(\int D(s, a)\, \mathrm{d}a\right) + H(\pi_E(\cdot \mid s)) + \left(\int D(s, a)\, \mathrm{d}a\right) - \left(\int \pi_E(a \mid s)\, \mathrm{d}a\right) \tag{6}$$

$$= H(\pi_E(\cdot \mid s)) - 1. \tag{7}$$

where $H$ is the entropy. (5) holds since $\log x \leq x - 1$ for all $x > 0$, and only equates when $x = 1$, i.e., $\hat{D}(s, a) = \pi_E(a \mid s)$. Since (7) is a constant, the maximum of $\hat{J}(D)$ can be taken when (5) equates, which means the optimal solution of $\hat{J}(D)$ is $\hat{D}(s, a) = \pi_E(a \mid s)$. Our KL penalty then is equivalent to using an IRL reward of $\log(\hat{D}(s, a)) = \log \pi_E(a \mid s)$.

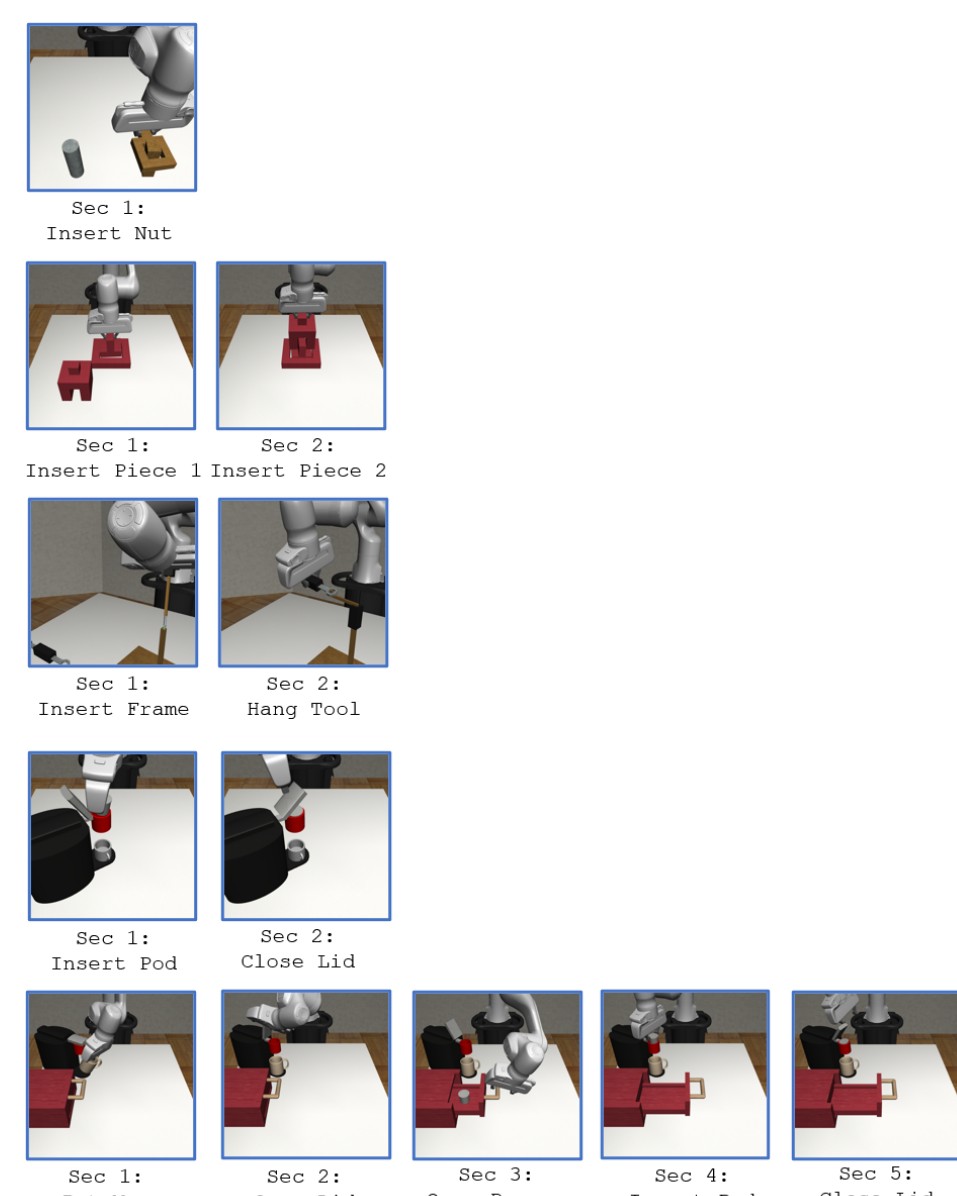

**Figure 7:** Handoffs per task, from top-to-bottom: *Square*, *Three Piece*, *Tool Hang*, *Coffee*, *Coffee Preparation*. In practice, we merge section 1 & 2, section 4 & 5 in *Coffee* and *Coffee Preparation*, as there are no TAMP actions in between.

## E   Tasks

**Square** and **Square Broad**.  The robot must pick up a nut and place it onto a peg.  This task has 1 handoff section, where the learned agent places the nut.  The **Broad** version increases the initialization range of both the nut and the peg.

**Three Piece** and **Three Piece Broad**. The robot must insert one piece into a base and place another piece on top of the first.  This task has 2 handoff sections, where the learned agent places the two pieces. The **Broad** version increases the initialization range of all three pieces including the base.

**Tool Hang** and **Tool Hang Broad**.  The robot must first insert a L-shaped piece into a base to assemble a frame, then hang a wrench off of the frame. This task has 2 handoff sections, where the learned agent inserts the L-shaped piece and hangs the wrench.  The **Broad** version increases the initialization range of all three pieces (base, L-shaped hook, and wrench).

**Coffee** and **Coffee Broad**. The robot must pick up a coffee pod, insert it into a coffee machine, and close the lid. This task has 1 handoff section where the learned agent inserts the pod and closes the lid. The **Broad** version increases the initialization range of the pod and the coffee machine.

**Coffee Preparation**. This is an extended version of **Coffee**. The robot must place a mug onto the coffee machine, open the lid, open the drawer with the coffee pod, pick up the pod, insert the pod into the coffee machine, and close the lid. This task has 3 handoff sections where the learned agent (1) places the mug and opens the lid, (2) opens the drawer, and (3) inserts the pod and closes the lid.

See Figure 7 for an illustration of all the handoff sections.

# F  Variance Across Seeds

In Figure 3, we show the best run out of 5 seeds. Here we provide the mean and standard deviation of the success rates in Table 3. We observe that although SPIRE still outperforms BC in terms of mean success rate in most of the tasks, our method exhibits unusually high variances in some of the tasks, for example, *Coffee*, *Three Piece*, and *Tool Hang*. In those tasks, one or more runs result in a performance significantly lower than the rest. Specifically,

- In *Coffee*, one run has 40% success rate, while the rest are all 100%;
- In *Three Piece*, one run has 22% success rate, while the rest are at least 98%;
- In *Tool Hang*, one run has 0% success rate and one has 6%, while the rest are at least 82%.

**Table 3:** Mean and standard deviation (in parenthesis) of success rates out of 5 seeds.

| Task | BC | RL [15] | Ours |
|---|---|---|---|
| Square | 92.4 (5.5) | 83.6 (36.7) | 99.2 (1.8) |
| Square Broad | 96.4 (4.1) | 100.0 (0.0) | 96.4 (5.4) |
| Coffee | 96.8 (4.1) | 40.0 (52.1) | 88.0 (26.8) |
| Coffee Broad | 41.6 (6.7) | 23.2 (12.1) | 84.4 (8.3) |
| Three Piece | 63.6 (6.7) | 0.0 (0.0) | 84.0 (34.7) |
| Three Piece Broad | 25.2 (7.7) | 0.0 (0.0) | 78.4 (5.0) |
| Tool Hang | 9.2 (4.6) | 0.0 (0.0) | 54.0 (46.8) |

Reinforcement learning methods are known to have high variances, especially in sparse reward settings. SPIRE partially alleviates this problem by enforcing the KL penalty for deviating from an anchor policy. However, in practice, such deviation can still happen.

Figure 8 compares the training curve of a successful run (with 88% final success rate) and a failed run (with 0% final success rate). The policy in the failed run drastically deviated from the BC policy early on in the training. This is likely related to the unusually large policy gradient loss, which the KL penalty term was unable to match and failed to constrain the policy.

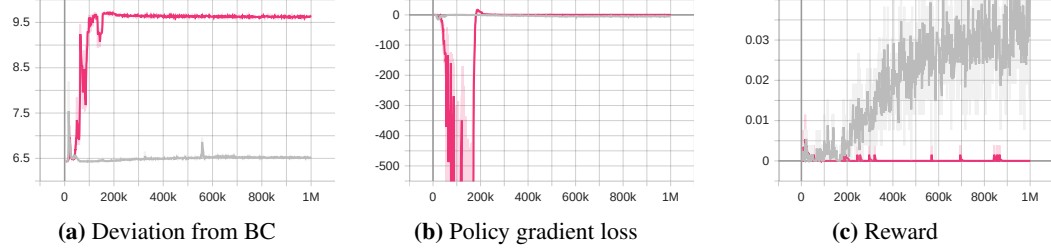

**(a)** Deviation from BC      **(b)** Policy gradient loss      **(c)** Reward

**Figure 8:** Comparing the (a) *Deviation from BC*, (b) *policy gradient loss*, and (c) *reward* training curves of a successful run (marked as grey) and a failed run (marked as red) in *Tool Hang*.

In our experiments, such an abrupt decrease in policy gradient loss happens frequently, with varying scales and timing, causing the training results to have high variance. Using an adaptive weight of the KL penalty might be a potential solution, which we wish to investigate in future work.

We do not believe 5 seeds are enough to quantitatively reflect the chance of such sudden deviation happening. An alternative solution would be to compare only the results where such deviation did not happen, which is why we chose to report the top-1 performing seed in our main paper.

# G TAMP Formulation

We specify TAMP formulations in manner similar to that of Mandlekar et al. [14]. We'll use the "Tool Hang" task in Figure 2 as an example of integrating traditional and learned actions within TAMP. The goal of this task is to pick and insert the frame into the stand and then pick and hang the wrench on the frame. Accordingly, a sequence of discrete action types that accomplishes this goal is:

$$\vec{a} = [\texttt{move}(...), \texttt{pick}(...), \texttt{move}(...), \texttt{insert}(...), \texttt{move}(...), \texttt{pick}(...), \texttt{move}(...), \texttt{hang}(...)].$$

We wish to learn the most contact-rich actions, namely `insert` and `hang`. For brevity, we'll describe just the `insert` action since the `hang` action has a similar description.

Like in [58], we represent planning state variables using *predicates*. Planning actions can be applied to planning states if they satisfy an action's of predicate *preconditions* (**pre:**). Upon application, an action modifies the truth value of the state variables through its predicate *effects* (**eff:**). Predicates and actions are parameterized by a list of typed arguments. Our TAMP domain involves the following types:

- `conf` - a robot configuration,
- `traj` - a robot trajectory comprised of a sequence of configurations,
- `obj` - a manipulable object,
- `grasp` - an object grasp pose,
- `pose` - an object placement pose,
- `policy` - a learned closed-loop robot policy,

and the following predicates:

- `AtConf`($q$: `conf`) - the robot is currently at configuration $q$,
- `HandEmpty`() - the robot's hand is currently empty,
- `AtPose`($o$: `obj`, $p$: `pose`) - object $o$ is currently at placement pose $p$,
- `AtGrasp`($o$: `obj`, $g$: `grasp`) - object $o$ is currently grasped with grasp pose $g$,
- `Inserted`($o$: `obj`; $\pi$: `policy`) - object $o$ is inserted into the stand as a result of executing policy $\pi$,
- `Motion`($q_1$: `conf`, $\tau$: `traj`, $q_2$: `conf`) - $\tau$ is a trajectory that connects configurations $q_1$ and $q_2$,
- `Kin`($q$: `conf`, $o$: `obj`, $g$: `grasp`, $p$: `pose`) - configuration $q$ satisfies a kinematics constraint with placement pose $p$ when object $o$ is grasped with grasp pose $g$,
- `PreInsert`($o$: `obj`, $g$: `grasp`, $q$: `conf`, $\pi$: `policy`) - object $o$ grasped with grasp pose $g$ and the robot at configuration $q$ are initiation states for policy $\pi$.

The `move` action is deployed using a joint-trajectory controller. Its parameters are the current robot configuration $q_1$, new configuration $q_2$, and trajectory $\tau$.

```
move(q₁ : conf, q₂ : conf, τ : traj)
  pre: {Motion(q₁, τ, q₂), AtConf(q₁)}
  eff: {AtConf(q₂), ¬AtConf(q₁)}
```

The `pick` action instantaneously grasps object $o$. Its parameters are object $o$, future grasp pose $o$, current placement pose $p$, and current robot configuration $q$.

```
pick(o : obj, g : grasp, p : pose, q : conf)
  pre: {Kin(q, o, g, p), AtPose(o, p), HandEmpty(), AtConf(q)}
  eff: {AtGrasp(o, g), ¬AtPose(o, p), ¬HandEmpty()}
```

The `insert` action is deployed using a learned policy $\pi$. Its parameters are object $o$, current object grasp $g$, current robot configuration $q_1$, future configuration $\widehat{q_2}$, and learned policy $\pi$. The $\texttt{PreInsert}(o, g, q_1, \pi)$ precondition models the set of combined object and robot states that initiate the action, thus defining the initial state distribution of the MDP for policy $\pi$. The $\texttt{Inserted}(o, \pi)$ effect models the set of terminal states of the MDP for policy $\pi$, namely where the sparse reward $r(s) = 1$. Unlike `move` and `pick`, `insert` is a *stochastic action* because end configuration $\widehat{q_2}$ depends on the execution of policy $\pi$. Because of this, we replan after executing each learned action.

$\texttt{insert}(o : \texttt{obj},\ g : \texttt{grasp},\ q_1 : \texttt{conf},\ \widehat{q_2} : \texttt{conf},\ \pi : \texttt{policy})$
  **pre**: $\{\underline{\texttt{PreInsert}(o, g, q_1, \pi)},\ \texttt{AtGrasp}(o, g),\ \texttt{AtConf}(q_1)\}$
  **eff**: $\{\underline{\texttt{Inserted}(o, \pi)},\ \texttt{HandEmpty}(),\ \underline{\texttt{AtConf}(\widehat{q_2})},\ \neg\texttt{AtGrasp}(o, g),\ \neg\texttt{AtConf}(q_1)\}$

**How do we select the handoff actions?**

SPIRE offers flexibility to human modelers for deciding which skills should be learned, depending on the precision requirements of the interaction. In our experiments, we used our understanding of the limits of the TAMP system to determine which skills should be learned. The hand-crafted skills, or the classical skills in our work include moving without colliding (transit and transfer motion) and grasping, which are fairly easy to automate. The rest of the skills are learned.

## H Bridging TAMP Planner and RL

In this section, we describe how we bridge the gap between a TAMP planner and RL in full detail. We follow our main text and use $\mathcal{S}$ and $\mathcal{A}$ to denote the state and action space respectively, with $T : \mathcal{S} \times \mathcal{A} \to \mathcal{S}$ being the state transition function and $S_* \subseteq \mathcal{S}$ being the set of goal states.

### H.1 Planner Formulation

We use TAMP as our default planner in our main text. In this section, we generalize our method to any planner with the following assumptions.

A SPIRE-*compatible* planner P, conditioned on a generated high-level plan, can be defined as a tuple $(N, S_{init}, \{S^i_{pre}\}, \{S^i_{eff}\})^N_{i=1}$, where $N$ is the number of handoff sections; $S_{init} \subseteq \mathcal{S}$ is set of the admissible initial states; for handoff section $i$, $S^i_{pre} \subseteq \mathcal{S}$ is the set of states where the preconditions for this section are satisfied, while $S^i_{eff} \subseteq \mathcal{S}$ is the set where the desired effects are realized. The planner works by iterating through the sections from $1$ to $N$. The planner should satisfy the following properties.

**Sequence validity.** For each section $i$, if the desired effects of the previous section are satisfied, i.e., the current environment state $s \in S^{i-1}_{eff}$ (let $S^0_{eff} = S_{init}$ for notational convenience), the planner then plans a sequence of actions that always terminates and leads to a new state that satisfies the preconditions of the current section, i.e., $s' \in S^i_{pre}$.

**Section validity.** For each section $i$, starting from any state $s \in S^i_{pre}$, a state $s' \in S^i_{eff}$ is reachable within a finite number of steps.

**Goal validity.** Any state that satisfies the effects of the final section should be a goal state, i.e., $S^N_{eff} \subseteq S_*$.

### H.2 Connecting the Planner with RL

SPIRE runs by interleavingly executing the planner P and the agent $\pi$, iterating through the $N$ handoff sections. In the $i$-th section, if the current state $s \notin S^{i-1}_{eff}$, then we terminate the process and report failure; otherwise, we execute the planner until the preconditions of this section are met. The planner then hands it off to the agent until the desired effects are met or a specified step limit has been reached.

**Theorem 1.** *Let* P *be a* SPIRE-*compatible planner, i.e.,* P *satisfies sequence validity, section validity, and goal validity. There exists an agent* $\pi_*$ *with which* SPIRE *reaches the goal state deterministically within a finite number of steps if the initial state is admissible by* P*, i.e.,* $s_0 \in S_{init}$ *and with a large enough step limit.*

**Proof.** With an induction on the handoff sections, we can conclude that given *sequence validity* and *section validity* there exists an agent $\pi_*$ with which SPIRE always reaches a state in $S^N_{eff}$ within a finite number of steps in each section. Given *goal validity*, the final state is always a goal state. $\square$

**Planner-induced MDPs.** In Sec 3.1, we formulate the planner-induced MDPs as a series of $N$ MDPs with shared dynamics and different reward functions and initial state distributions, each corresponding to a planner handoff section. We can now formally define each MDP's reward function: For the $i$-th MDP, the reward function is $r^i(s) := \mathbb{1}_{\{s \in S^i_{eff}\}}$. The support of the initial state distribution satisfies $\mathrm{supp}(p^i_0) \subseteq S^i_{pre}$.

**Sample collection rate issue.** One critical issue we encountered when integrating a TAMP planner into RL training was the dropped rate of sample collection. To give more context, suppose it takes a fixed time of $t$ to sample one frame of environment interaction with an RL agent. Under this assumption, the natural sampling speed with pure RL is $1/t$ frames per second (FPS). When combined with a TAMP planner, let the planning time before each handoff section be $T$, and let the RL agent run for at least $H$ steps in each section, the resulting sampling speed becomes at most $H/(T + tH)$

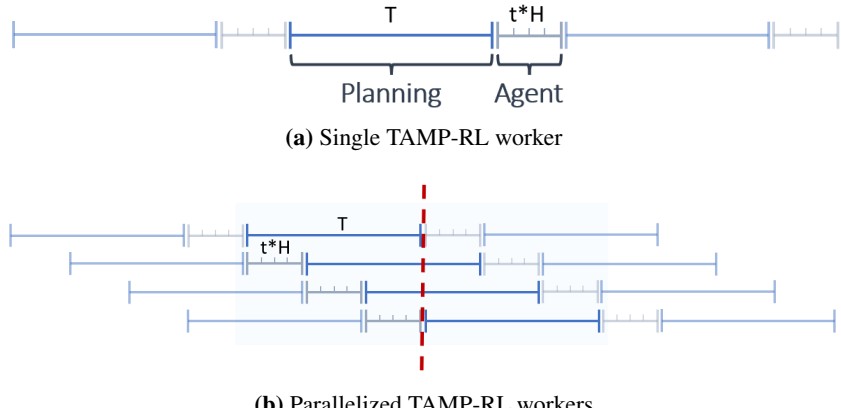

**(a)** Single TAMP-RL worker

**(b)** Parallelized TAMP-RL workers

**Figure 9:** Demonstrations of a single-process TAMP-RL pipeline and multi-process TAMP-RL pipeline.

FPS (see Figure 9a), a relative slowdown of $Ht/T$. In practice, the TAMP planning time $T$ can be much longer than $tH$, making this slowdown significant.

**Initial state distribution issue.** We know that the support of the initial state distribution in the $i$-th section is within $S^i_{eff}$. However, even if we assume the planner is fixed, the distribution can still change based on the agent behaviors in the previous sections. In the case of training a single agent for all sections, the distribution is even more prone to policy changes as the completion rates of previous sections can affect the appearance rates of later sections.

### H.3  Multi-Worker Scheduling Framework

To resolve the two aforementioned issues, we designed the *multi-worker scheduling framework* that utilizes parallelization, as introduced in Sec 3.5.

**Solving the sample collection rate issue.** See Figure 9b. The multi-worker system runs $n$ TAMP planners in parallel. When a worker finishes planning and reaches the next handoff section, it will notify the scheduler to be available and enter hiatus, until selected by the scheduler for RL interactions. If the system has enough workers, specifically when $n \geq \frac{T}{tH}$, there always will be a TAMP worker available for RL and reach the optimal sample collection rate of $1/t$ FPS.

**Solving the initial state distribution issue.** We attempt to alleviate the changing initial state distribution issue by applying a selection strategy to the scheduler. Specifically, the `sequential` strategy only accepts a section when the success rate of passing all the previous sections reaches a threshold, which indicates that the agent's behavior in previous sections becomes stable. The framework allows for other potentially more sophisticated selection strategies, even though we do not find it necessary in the scope of our work.

# I Ablation: SPIRE without Multi-Worker

We provide an additional ablation study on how the sample collection rate boost from the multi-worker system benefits the overall performance. We run SPIRE with 16 workers and 1 worker respectively for the same amount of 6-hour wall-clock time, for the first insertion of the *Tool Hang* task. The result is shown in Table 4.

**Table 4:** Comparing the success rates of the first insertion in *Tool Hang* with and without multiple workers.

| # Workers | Success rate | Sampling FPS |
|-----------|-------------|--------------|
| 16        | 98%         | 55.4         |
| 1         | 42%         | 7.7          |

## J   Additional Experiment Results

We show the RL learning curves in Figure 10 comparing SPIRE and naive RL. The x-axis represents the environment steps and the y-axis represents the success rate.

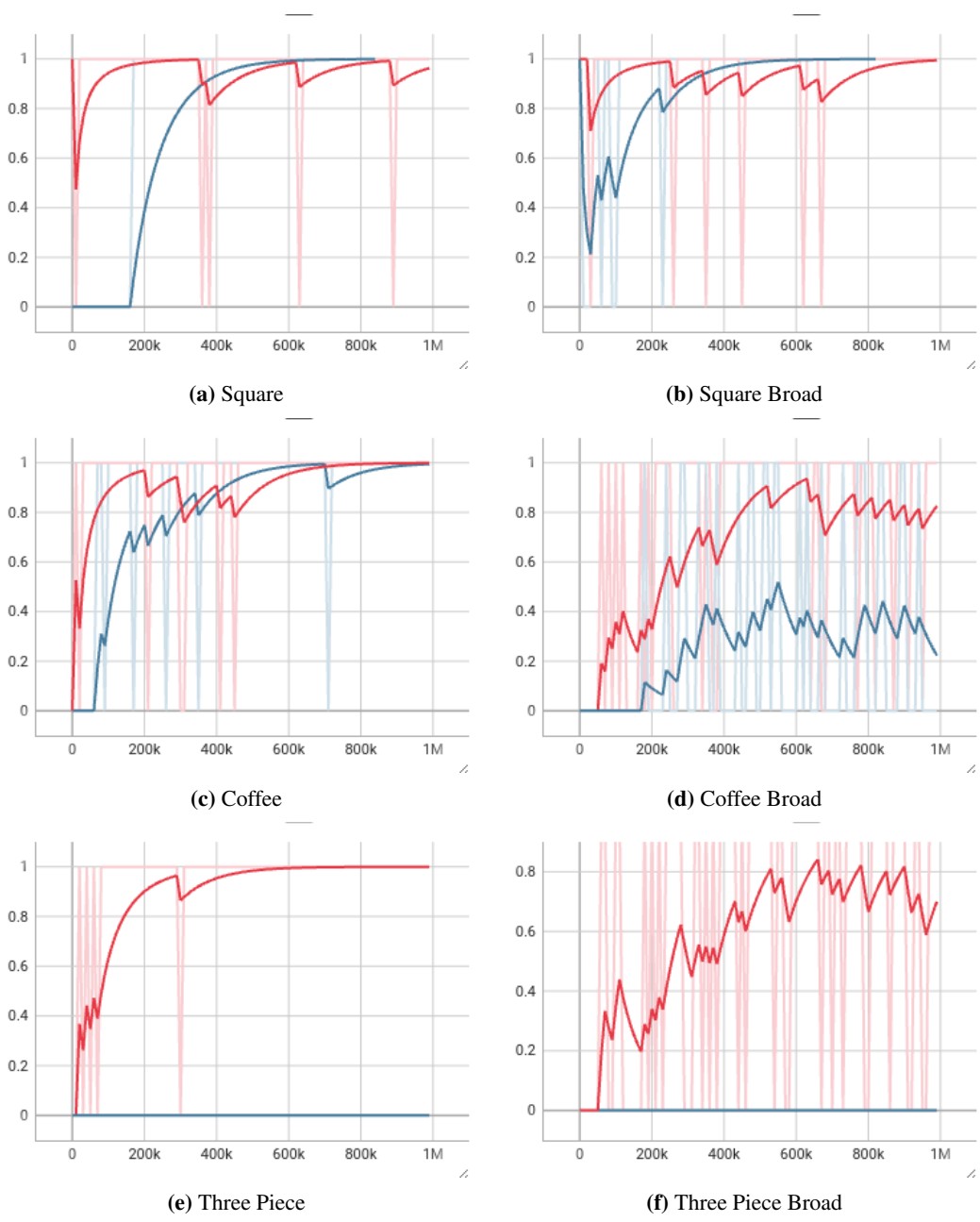

(a) Square

(b) Square Broad

(c) Coffee

(d) Coffee Broad

(e) Three Piece

(f) Three Piece Broad

**Figure 10:** Comparing the RL learning curves of SPIRE and naive RL.

