# OpenReview forum: "SPIRE: Synergistic Planning, Imitation, and Reinforcement Learning for Long-Horizon Manipulation"
_robot-learning.org/CoRL/2024/Conference — CoRL 2024_

### Official Review · Reviewer_doj6 · 2024-07-18
**Major paper organization issues obscure interesting and useful ideas about BC-initialized RL skills in TAMP system**

**Originality:** 3
**Technical Quality:** 4
**Clarity Of Presentation:** 2
**Potential Impact:** 3
**Recommendation:** 2
**Confidence:** 4

**Review:**

**Clarity:**
The main contribution of the paper is hindered by the paper presentation and organization.
Organization seems easier to fix: Appendix G and H contain information that seems way more important to the main contribution than the RL finetuning techniques described in Section 3.4. Much of the section 3.4 content seems better suited to the appendix and content in Appendix G and H seems more suited to be in the main paper.
For presentation, I see two consistent issues. The first is excessively vague language for important technical information. The second is presenting less important information with equal or higher emphasis than more important information.

Some key examples from contributions (62-73), which really should be the clearest writing in the paper:

Less important information hiding more important info:  On line 65, the second contribution lists warm-starting with BC (well-explored) and then mentions the multi-worker TAMP framework, which is more novel.

Excessively vague language: the first contribution states that the approach is "synergistic" and then states 3 components that were combined. "Synergy" is a low information-density word. Thus, the  contribution is missing a claim on how it integrates these components in a new way.  An improved first contribution can say something stronger about what the synergy is and what it does.

Another example of excessively vague knowledge is in lines 132-137 of the problem formulation. The authors provided clarifications in the comments, which are also addressed in the appendix, but the main paper describes it as "TAMP determines."



**Originality and significance:**

~~The first and presumably main contribution seems to be a straightforward and minor modification of [14] by replacing BC policies with BC-initialized RL policies. [15] already does RL policies integrated into a higher-level planning system~~
- The paper addresses the challenge of sampling valid initial states for RL policies that need a large amount of data. Bootstrapping with BC enables safer and more efficient exploration.

- The ideas presented related to initializing an RL policy with an IL policy is not a “key insight” introduced by this paper, which seems like an overclaim. The KL divergence term and initialization of some weights from a BC policy are known to the authors, which they mentioned in the “RL with experts” section of related work and even show a relationship between their method and [47]. The distinction of prior work being “focused on low-dimensional or single-stage MDP settings” seems insignificant given Huang et al. 2022 and Liu et al. 2022 use image inputs in a driving environment

**Strengths:**
- Multi-worker parallelization makes RL feasible when running the planner to collect in-distribution data.
- RL finetuning does greatly improve the quality of plans executed with HITL-RL
- ~~Paper is generally well organized: information is in the expected sections. It was easy to navigate and find more details.~~
- ~~Contributions are stated clearly~~
- Experiments measure the contributions
-Warmstarting with an IL policy solves the task in a "human"-like way. As authors note, this can be good for safety by generally doing the motion in a "sensible" way
- Contains a robot interacting with the objects in an experimental setting that is practical to exist in the real-world
- As far as insertion and hanging type problem settings go, these are relatively difficult due to the unwieldy shapes of the objects.
- Very nice results in the E2E evaluation
    - contains some (easier) problems where baselines perform the same as the method, which reflects a good-faith implementation of the baselines
    - Significantly improved metrics from BC/RL in several tasks at the bottom of Figure 3.
        - success rate
        - episode length
        - number of needed human demonstrations
- Compared against strong baselines.
- Videos give qualitative information about what the policies look like
- Limitations section presents assumptions about the system that seem like the most important to me as well.

**Weaknesses**
- The main weakness is in presentation and organization. More information is in "clarity"
- The first RW paragraph seems focused on HiRL and doesn’t include a discussion comparing with other planning approaches with learned skills.
- Evaluation appears to be entirely in simulation, which makes the safety argument in Figure 4. slightly less compelling. It also shows less potential for dealing with noise/occlusion that happens in the real world but not in simulation.
- The skills used for “handoff” such as insert and hang don’t seem compellling to me as not suitable for motion planning in this domain, especially since these tasks can be quite easy to do kinematically or with visual servoing, especially in manufacturing settings with multiple available camera views.
    - Contact-rich does not imply difficult to specify classically. Contacts can be used to make insertion easier, such as by using some compliance (Mason 1981)
    - If the same exact mechanisms are used as in [14], please say that explicitly. It would also make it clearer what this work does that [14] does not do beyond adding an RL fine-tuning step.
    - What are the "classical" skills supplied? How are they specified?
- The curriculum learning ablation study seems very specific to that skill and not insightful more broadly.
- Labels in figures do not describe the action represented by the section despite the fact that based on my understanding of [14], those actions are grounded in a specific action with specific preconditions, postconditions, etc… Figure 1 and 7 would be easily improved with such labels. Potentially Figure 4 as well.


**Typos:**
- "supplematary" folder is misspelled in the supplementary material
- a contact-rich skill is called “hand” in the caption of Figure 2 and is probably supposed to say “hang”

**References**

- Huang, Zhiyu, Jingda Wu, and Chen Lv. "Efficient deep reinforcement learning with imitative expert priors for autonomous driving." *IEEE Transactions on Neural Networks and Learning Systems* 34.10 (2022): 7391-7403
- Liu, Haochen, et al. "Improved deep reinforcement learning with expert demonstrations for urban autonomous driving." *2022 IEEE intelligent vehicles symposium (IV)*. IEEE, 2022.
- Mason, Matthew T. "Compliance and force control for computer controlled manipulators." *IEEE Transactions on Systems, Man, and Cybernetics* 11.6 (1981): 418-432.

**Quality Of The Limitations Section:**

3

**Questions For Rebuttal:**

1. Please clarify the significance of adding IL initialized / RL fine-tuned policies that is not shown by [14] or [15] What are the challenges and added complexities of finetuning IL policies to be used in a TAMP system that do not apply to BC policies or RL policies in general?


2. What TAMP method do you use? At the very least, what is needed from a TAMP system to be compatible with this method? It seems to me implicitly that the TAMP method needs to be [14]. Ex) how are goals specified, how is state information used. A PDDL file in the supplementary would help.

3. Related to above, where do the postconditions come from and how are they used during both IL and RL? Does a demonstration need to satisfy the postcondition? How does the demonstrator know what to do?

4. If you have the preconditions and postconditions and a simulated environment, why do you need to run the TAMP planner to gather the dataset for learned skills?

**Robotics Focus:**

3

**Summary Of Paper:**

The key idea of this paper is that combining behavior cloning (BC) and RL fine-tuning can lead to completing tasks more efficiently and reliably within a hybrid TAMP system that uses a combination of learned and non-learned skills. Authors state a secondary contribution of suggested techniques for efficient RL finetuning from a BC prior. Experiments show large improvements in success rates (reliability), episode length (efficiency) and demo data-efficiency.

**Summary Of Recommendation:**

Currently, the (main) paper does not clearly communicate the authors ideas about integrating BC-initialized RL policies into their TAMP system. After reading author responses, I can see why the BC/IL addition is a nontrivial and helpful addition to HITL-TAMP. My scores are updated to reflect a better understanding of the technical contribution. I am leaving my recommendation as "Weak Reject" since I think the paper needs major revision beyond the revised manuscript.  I recommend to the authors that they re-organize the paper to highlight the most significant ideas in the main paper.

---

### Official Review · Reviewer_TuDy · 2024-07-21

**Originality:** 4
**Technical Quality:** 4
**Clarity Of Presentation:** 3
**Potential Impact:** 3
**Recommendation:** 4
**Confidence:** 3

**Review:**

Strength
- The proposed idea leverages the complementary strengths of TAMP, IL, and RL.
- The quantitative evaluation and the ablation studies well support the design choices of the framework.
- This work demonstrates the successful training of long-horizon tasks with a handful of human demonstrations.
- This paper describes its limitations and the assumptions on the evaluation tasks well.

Weakness
- It would be good to describe the limitation of previous works on TAMP in the Related Works.
- I might have missed the details, but it does not seem to describe how TAMP prompts handoff sections. Additionally, it would be helpful to elaborate on what actions can be considered 'too hard to plan' by the TAMP planner in more precise terms. Are these tasks contact-rich, or do they involve precise manipulation? Why are they difficult to plan by the TAMP planner?
- The paper states that SPIRE does not introduce desirable behavior. However, it seems to be applied only to handoff sections. TAMP may not guarantee desirable actions.

**Quality Of The Limitations Section:**

3

**Questions For Rebuttal:**

- The proposed method uses TAMP. Therefore, it would be worthwhile to add a vanilla TAMP baseline that does not use the gated RL/BC policy. This would highlight the strength of SPIRE over the TAMP planner alone. If this is not available, is there any way to adapt it for a baseline?
- Is the TAMP used in SPIRE model-based? If so, it might require full geometric information of the environment. However, it seems that the observation space is provided as visual observation from multiple viewpoints. Doesn't this cause partial observability for TAMP?

**Robotics Focus:**

3

**Summary Of Paper:**

This paper presents a hybrid learning method aimed at long-horizon manipulation tasks. Within the framework, the TAMP layer executes manipulation for plannable actions. For non-plannable actions, it enters handoff sections. For training, human demonstration data is collected in a human-in-the-loop manner in the handoff sections. Then, policies are initially trained on the demonstration data through behavior cloning and later fine-tuned by RL. During testing, the TAMP planner and these handoff policies are used for manipulation tasks.

**Summary Of Recommendation:**

I think the idea of integrating TAMP, IL, and RL in this paper is interesting. Additionally, the proposed framework is sound, and the results support the strength of the method. However, there are some unclear parts regarding SPIRE’s TAMP layer, as mentioned in my review.

---

### Official Review · Reviewer_bC4r · 2024-07-27

**Originality:** 3
**Technical Quality:** 4
**Clarity Of Presentation:** 4
**Potential Impact:** 3
**Recommendation:** 3
**Confidence:** 5

**Review:**

Authors present a framework that is named SPIRE, that relies on TAMP for decomposing the manipulation problem into a sequence of individual skills, similar to HTIL-TAMP. TAMP hands over the control of the robot to human user/RL. Skills are learned with combination of BC and RL. BC is not directly used for warmstarting but for constraining RL policy with divergence loss. Seems that not all the skills are learned but only mentions a hand are in some problem domains, although they are long-horizon.

Strengths:

- Parallelisation, not sure if this is also in the HITL-TAMP, Multiworker.
- Relatively good performance compared to baseline.
- Comparison to two related similar works: HITL-TAMP, TAPM + BC and plan-sequence-learn + TAMP + RL.
- The curriculum approach is interesting.

Weaknesses:

- BC and RL combination is not new, many examples in [1], chapter RL with HiL.
- Many assumptions seem to be hidden, e.g. segmentation, TAMP model, "classical skills", etc. They are also tackled in the questions.

minor:
- In the abstract, Robot Learning is not a technique.
- In the abstract, problem complexity is not proportional to the horizon length, it is exponential.
- Line 27: It is not only teleoperation, it can also be kinesthetic teaching, e.g., [1].
- Citations [7] and [13] are misinterpreted.
    - [13] is not tackling long-horizon problems, it is focused on OOD generalization.
    - [7] is temporal difference learning, and not long-horizon.
- Many citations for some concepts without detailed explanation why [1, 2, 3, 4, 5], it is overwhelming to read and include very general works, more like a survey, but also shallow in that regard. Perhaps in those cases better to refer to some survey if some specific work is not needed.
- Line 85, incomplete sentence.
- Line 93, "learning robot..."
- Line 121, alternating TAMP?
- Line 122, why call hand-off?

[1]Celemin, Carlos, et al. "Interactive imitation learning in robotics: A survey." Foundations and Trends® in Robotics 10.1-2 (2022): 1-197.

**Quality Of The Limitations Section:**

2

**Questions For Rebuttal:**

Questions:

- Which skills are learned and which are hand-crafted? What are "classical skills"?
- Where does the TAMP model come from?
- How is it decided which tasks are handed over?
- How are symbolic predicates obtained?
- Which planner is used?
- In the pipeline for handing over, is the complete plan from TAMP available?
- TAMP planner seems to use cover part but it is not explained.
- Why is RL failing on those domains?
- How does your method generalize to novel domains?
- Learning curve for RL and SPIRE would be very informative.

**Robotics Focus:**

3

**Summary Of Paper:**

A framework based on TAMP utilizing BC and RL.

**Summary Of Recommendation:**

Good paper, but nothing really outstanding to be accepted. Seems very incremental with many assumtions that need to be discussed.

---

### Author Rebuttal · Authors · 2024-08-10

The rebuttal file includes our revised manuscript, with changed text marked in blue.

---

### Decision · Program_Chairs · 2024-09-04

**Decision:**

Accept

**Comment:**

This paper combines TAMP, behavior cloning, and RL, to built a system that can solve complex long-horizon manipulation problems.

Although on the first sight this work can be seen as a mere combination of TAMP, BC, and RL, this paper is relevant for the community because it bridges the gap between TAMP that often cannot deal with manipulation tasks that require high precision, and RL which struggles with long horizon tasks. This combination and the insights are novel, and hence of value for the community.

The authors provided a strong rebuttal addressing most of the concerns of the reviewers.

Please note that according to one reviewer, the paper still need further major revision for clarity and presentation.